

# An ethical visualization of the NorthCOVID-19 model

Andrew Fisher, Neelkumar Patel, Preetkumar Patel, Pruthvi Patel, Vinit Krishnankutty, Vaibhav Bhat, Parth Valani, Vijay Mago and Abhijit Rao

Department of Computer Science, Lakehead University, Thunder Bay, Ontario, Canada

## ABSTRACT

When modelling epidemics, the outputs and techniques used may be hard for the general public to understand. This can cause fear mongering and confusion on how to interpret the predictions provided by these models. This article proposes a solution for such a model that was created by a Canadian institute for COVID-19 in their region; namely, the NorthCOVID-19 model. In taking these ethical concerns into consideration, first the web interface of this model is analyzed to see how it may be difficult for a user without a strong mathematical background to understand how to use it. Second, a system is developed that takes this model's outputs as an input and produces a video summarization with an auto-generated audio to address the complexity of the interface, while ensuring that the end user is able to understand the important information produced by this model. A survey conducted on this proposed output asked participants, on a scale of 1 to 5, whether they strongly disagreed (1) or strongly agreed (5) with statements regarding the output of the proposed method. The results showed that the audio in the output was helpful in understanding the results (80% responded with 4 or 5) and that it helped improve overallcomprehension of the model (85% responded with 4 or 5). For the analysis of the NorthCOVID-19 interface, a System Usability Scale (SUS) survey was performed where itreceived a scoring of 70.94 which is slightly above the average of 68.

## INTRODUCTION

The spread of COVID-19 has had a great impact around the world and was classified as a global pandemic in March 2020 as a result (*World Health Organization, 2020*). The exponential rate at which this disease spread has resulted in an abundance of research into how populations can try to handle the disease and do their part in preventing it's transmission (*World Health Organization, 2020*; *Remuzzi & Remuzzi, 2020*; *Wong et al., 2020*). In the past, the idea of modelling these types of diseases has been a prominent area of interest in the mathematical modeling field as it can assist in future planning and preparation (*Roddam, 2001*; *Angstmann, Henry & McGann, 2016*; *Garibaldi, Moen & Pissarides, 2020*). A basic–yet very popular–type of modelling technique known as the **S**usceptible **I**nfectious **R**ecovered (SIR) model (*Kermack & McKendrick, 1927*) has been used in the past to model diseases such as Ebola (*Berge et al., 2017*), influenza (*Osthus et al., 2017*), and measles (*Bjørnstad, Finkenstädt & Grenfell, 2002*) for example. However, the

Corresponding author
Andrew Fisher, afisher3@lakeheadu.ca

outputs of these models might be difficult for the general public to understand as it can not be assumed that everyone will have the necessary mathematical background needed to properly interpret these results (*Pickering & Kara, 2017*). This is an important issue to address for two main reasons: (1) to prevent fear mongering (*Begley et al., 2007*) from not being able to fully understand how these models are developing their conclusions and (2) to help the population understand the impact of having appropriate community-level prevention measures for these diseases be followed. This work will look at the NorthCOVID-19 model that was created by a Canadian institute (*Savage et al., 2020*) for epidemic modeling in the North Ontario region. The authors of *Savage et al. (2020)* extended the SIR model (*Kermack & McKendrick, 1927*; *Angstmann, Henry & McGann, 2016*) to include a second population to run in parallel with the first, as well as additional states to assist in forecasting the hospital demand for the region which resulted in a total of 32 configurable parameters. Moreover, a web interface is publically available (https://covid.datalab.science/) where users can configure these parameters and run a simulation to see the results.

The layout of this article is as follows: it will start with a basic introduction to the SIR model, as well as the motivation behind assisting in public understanding. Next, a brief overview of the NorthCOVID-19 model will be provided, as well as an analysis of the current interface to see how difficult it may be to use. Next, our proposed video-generation-solution will be discussed that utilizes artificial intelligence (AI) to enhance explainability, along with frames from an example of the output to show the ethical decisions made throughout its development. We refer to this as an *ethical visualization* which, in the context of this work, is a visual representation of data that takes ethics into consideration to ensure that any user interpreting it is not left confused or with unanswered questions. A more in-depth explanation is presented in the issues section where we show specifically what was taken into consideration for this visualization. Lastly, the results of a survey presented to users outside of this project is shown, with the purpose being to gather their opinions of this proposed output and compare it to the currently available output on the web interface. The contributions of this work are as follows:

1. A method entirely motivated by ethics to visualize epidemic modelling
2. An application of AI for social good (*Inkpen et al., 2019*)
3. An analysis of a public interface for epidemic modelling from an ethical standpoint

In the context of epidemic modelling, it is important to ensure that the data–particularly for COVID-19 (*Roser et al., 2020*; *Hoseinpour Dehkordi et al., 2020*)—is represented using *ethical visualization* techniques because it is projecting the outcome of the entire population. Therefore, the output should be represented in a way that can not only be interpreted easily by any user but also visualized ethically.

## SIR model

The origin of the SIR model dates back to 1927 as a contribution to the mathematical theory of epidemics (*Kermack & McKendrick, 1927*). Since then, there have been many works that have expanded on the core concepts of the SIR model and have proven to be a simple

yet effective way to predict the course of various epidemics (*Roddam, 2001*; *Angstmann, Henry & McGann, 2016*; *Osthus et al., 2017*; *Garibaldi, Moen & Pissarides, 2020*). For the NorthCOVID-19 model (*Savage et al., 2020*), a time-series variation was used as the basis where the differential equations represent a population's change over a set time period (*Angstmann, Henry & McGann, 2016*; *Osthus et al., 2017*). In the core equations of the model, there are a total of four parameters; contact rate (the number of people an *infectious* individual can infect on average), infectivity rate (the percent chance a *susceptible* individual can become infected by an *infectious* individual), recovery rate (the percent of *infectious* individuals that recover from the infection after one time period), and the total population size (*Kermack & McKendrick, 1927*; *Roddam, 2001*). One time period, for this article's purpose, is a tenth of a day (*i.e.,* an update is made to the system every 0.1 days) as it is defined as such in the NorthCOVID-19 model.

## Public understanding

When discussing the results of these epidemic models with the general public, it may be difficult to convey the reasoning behind their outputs and why it is important to fully understand. To further the motivation behind this article's purpose, the ACM Code of Ethics (*Gotterbarn et al., 2017*) were reviewed and the following four codes were found to be the most relevant to this work:

- 1.3: Be honest and trustworthy
- 2.7: Foster public awareness and understanding of computing, related technologies, and their consequences
- 3.1: Ensure that the public good is the central concern during all professional computing work
- 3.2: Articulate, encourage acceptance of, and evaluate fulfillment of social responsibilities by members of the organization or group

With the output of the proposed method, a great deal of care is taken to ensure that no important information is missed or downplayed to the end-users (*Hoseinpour Dehkordi et al., 2020*). The goal is to help the general public understand the outputs from these types of models so that they can interpret it without any confusion.

## Prior works

When searching for related works in visualizing epidemic models we have found that, to the best of our knowledge, next to none exists in recent literature. Looking further back, however, work from *Höhle & Feldmann (2007)* in 2007 proposed a R(programming language) package that included visuals for "stochastic epidemic models". The authors did **not** incorporate ethics into the work as it simply displayed the raw data in a graphical form. In 2011, the authors of *Maciejewski et al. (2011)* presented a computer application called "PanViz" that was targeted towards public health officials in the USA to simulate a pandemic. The visualization aspect of this program showed graphs of the statistics and, most notably, how the disease may spread throughout various districts in the country-which was further detailed in another work by two of the authors (*Afzal, Maciejewski & Ebert, 2011*). In both cases, though, ethics was not taken into consideration and it was

not targeted towards the general public. When interpreting the outputs of an epidemic model, a prominent thought may be how *trustworthy* the results are. This is a valid concern as these models are trying to predict the *future* outcome of a given scenario. In recent literature, visualization of such "uncertainty" has been explored as it affects the trust and understanding of the individual interpreting it *Greis et al. (2017)*, *Hullman (2019)* and *Hofman, Goldstein & Hullman (2020)*. In the work presented by *Kale, Kay & Hullman (2019)*, the authors looked at the complexity that researchers face in managing such uncertainties in a way that does not discredit their work. They provide solutions such as showing all possible outcomes based on the uncertainty or simply disclosing the reason it exists in the first place (*Kale, Kay & Hullman, 2019*). In either case, it should not be too complex for the end-user to understand to prevent confusion or distrust (*Greis et al., 2017*). In the context of our specific work, a number of uncertainties exist when projecting for COVID-19; especially when explaining it to lay users. Such issues will be raised along with the solutions that were incorporated in the proposed output.

## MATERIALS & METHODS

### NorthCOVID-19

The NorthCOVID-19 model was created by the authors of *Savage et al. (2020)* as a variation of the SIR model that considered both urban and rural populations when simulating an epidemic. The foundation of this model uses sets of differential equations to move individuals from one state to another. It also takes into account resources reaching maximum capacity by having "overflow" equations to emulate the path individuals will take in that instance (*Savage et al., 2020*). For example, if an ICU has reached maximum capacity but an individual is set to enter it, they will, instead, go through these "overflow" paths. Furthermore, when compared to the SIR model, an additional five states (Hospitalized, ICU, ICU Discharge, Ward, and Death) have been added within two populations running in parallel because, for the target location of the Northern Ontario region in Canada, rural communities commonly access urban health services for more specialized care (*Savage et al., 2020*). However, for the purposes of this research, the proposed output to be discussed will only take **one** population into consideration; specifically, the **urban population** to improve the understanding of outputs for one location at a time without the added complexity of another population, thus affecting the results.

Another important aspect of this model is the **intermediate states** denoted as purple squares. These are sub-states where individuals need to be held for a number of time periods before travelling down the flow they are connected to. For example, in the "*ICU*" state, one of the parameters used is the number of days an individual will need to stay there before being moved to the "*Ward*". If an individual does not fall under the flow leading to the "*Death*" state, they will then be held in the intermediate state so that they are only affected by the flow leading to the "*Ward*" state while still occupying a spot within the "*ICU*" state. The main limitation of this model is that the parameters are entirely static in nature- which is different in reality with public health policies such as lockdowns, for example, which would change the population's average contact rate. Regardless, this model

is important as it is able to project what would happen if no changes were made to curb the spread of the modelled disease. Further details on the model and parameters can be found in *Savage et al. (2020)*.

### Current interface and output

When the authors of NorthCOVID-19 (*Savage et al., 2020*) made the model, a web interface was designed alongside it to provide researchers in the field an accessible means of interacting with it. This interface needed to accommodate a total of 32 parameters across two populations so that individual aspects of the model could be modified. Once this has been configured to the user's specifications, it can be submitted where the simulation is run in a matter of seconds before being returned to the user shortly after.

Once the results have been computed, the first thing displayed to the user is a summary of what occurred in the simulation. Second, the user is shown the number of *susceptible*, *infectious*, and *recovered* individuals per capita of 10,000 over time. Next, the user is shown the number of deaths per capita of 10,000 over time. Lastly, a graph showing the *ICU* usage over time is presented to the user, as well as a plot of how many care units would be in place, ideally, to accommodate everyone. The user is given the option to download a spreadsheet of the raw results from the simulation that shows the fractional, point-in-time counts, for each state, every 0.1 timesteps.

### Issues

To get an initial idea of how the NorthCOVID-19 interface was being perceived, we asked the authors (*Savage et al., 2020*) for their feedback on possible issues that may surface for the general public. It was noted by them that it had already been demonstrated to other researchers, public health officers, policy makers, and physicians so they had a good idea of how users were assessing their system (*Pearce et al., 2020*). Taking their points and observations into consideration, a few issues became apparent:

1. The initial interface of 32 parameters could be overwhelming to a user that does not understand how each parameter works
2. The raw results, being so granular and in decimal format, could be confusing to a user that does not understand the implementation of mathematics in the model
3. The graphs, although valuable to researchers in the field as the results have been generalized to per capita values, could be confusing to an end user that does not understand how these ratios work or what they mean to the actual population size

These issues can be further addressed by the ACM Code of Ethics (*Gotterbarn et al., 2017*) discussed before. For issue 1, codes 2.7 and 3.1 can be applied as we want to ensure that the users understand the reasoning behind these results. Although there are 32 parameters, the additions on top of the base SIR model come from the hospitalization states that have been added to NorthCOVID-19 (*Savage et al., 2020*). Since this aspect of the model would be specific to the regions it is being applied to, the general public would assume that these states have been configured to what their area expects. Therefore, in the proposed output, we only display parameters from the model that relate to the disease itself—contact rate, infectivity, and recovery time—as they would be most likely to vary

based on location-specific statistics. Additionally, however, the NorthCOVID-19 model also has parameters for how many individuals are initially infected as well as what the capacity limit of the intensive care unit (ICU) is *Savage et al. (2020)*. These parameters are also important for the general public to know as it shows how the disease begins in their region as well an important value showing the handling capacity of their hospitals. For issue 2, code 1.3 and 3.1 can be applied as we want ensure that the general public doesn't dismiss the results of these models by helping them understand results in a better format. Lastly, for issue 3, all of the codes discussed in the *Public Understanding* section can be applied as the output of the model should be understandable for all individuals since it has a great deal of importance to the general public. This can be accomplished by using the actual values output from the model rather than per-capita or ones that have been normalized.

## Proposed epidemic model output
### Video output
The first aspect to the proposed solution is a video output that is generated based on the following values made available by the NorthCOVID-19 model: the parameters used, summarized results, and raw data. In practice, these values would be based on local statistics that are relevant to the audience being targeted. To demonstrate the proposed method, arbitrary values will be used to give an idea of how the video output would look. In the very first frame, we display to the user the preset values for the important parameters identified. This frame stays up long enough for the audio output (discussed in the next section) to read all values out to the viewer- an example can be seen in Fig. 1. This is to ensure that the viewer understands the setup of this particular output before any of the results are presented.

In the next frame, the proposed solution first presents the raw *infectious* numbers over time. The scale on this graph is an absolute value instead of being a per capita value so it is much more transparent and easier to understand (*Kienzler, 1997*; *Kostelnick, 2008*)—an example of this is shown in Fig. 2. However, it does not convey another crucial design choice: the graph is animated to show the numbers over time from the first day to the last day. During the animation, the audio output discussed in the next section plays to walk the viewer through what happened as a result, with the total number fading in for the conclusion of this frame. The reasoning for this visualization setup is to ease the viewer into the results instead of abruptly showing a graph that they need to interpret by themselves (*Pickering & Kara, 2017*). The same design choice was made for the frames following this one as shown in Figs. 3 and 4.

Additionally, the ordering of these frames took ethical thinking into mind as we wanted to properly lead up the morbid results (*i.e., deaths*) by not displaying it first to the users. To do so, the video first shows the user how many individuals actually became infected along with how many were among the population at any point in time. This assists in leading into what the load was on the hospital and how the *ICU* may or may not have reached the maximum capacity over time. Then, the deaths are presented along with the *recovered* so the viewer is able to see both outcomes, together, on the same scale.

# The results in this video are based on the following parameters:

Initial population: 110,000

Initial infected: 10

ICU capacity: 40

Contact rate: 12.0

Infectivity: 2.6%

Illness duration: 7.0 days

**Figure 1** The first frame in the proposed solution highlighting the initial parameters used in this example simulation.

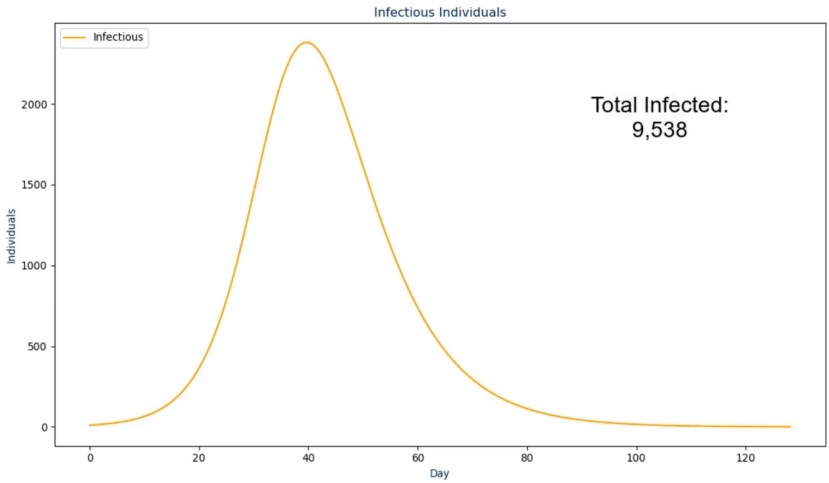

**Figure 2** The second frame in the proposed solution.

Once the last graph has been shown to the viewer, a concluding frame is shown where all peaks that were previously shown are presented, on the same scale, in a bar chart. As with the other slides and to be discussed in the next section, an audio output accompanies this to reiterate each of the results to the viewer. By displaying the peaks in this format, one can view the magnitude of each peak relative to one another with little to no confusion (*Pickering & Kara, 2017*). This is particularly important for the *infectious*, *recovered*, and *death* peaks as this will typically be what the general public is most concerned with *Kienzler (1997)* and *Begley et al. (2007)*. To conclude this section, the code for the proposed output

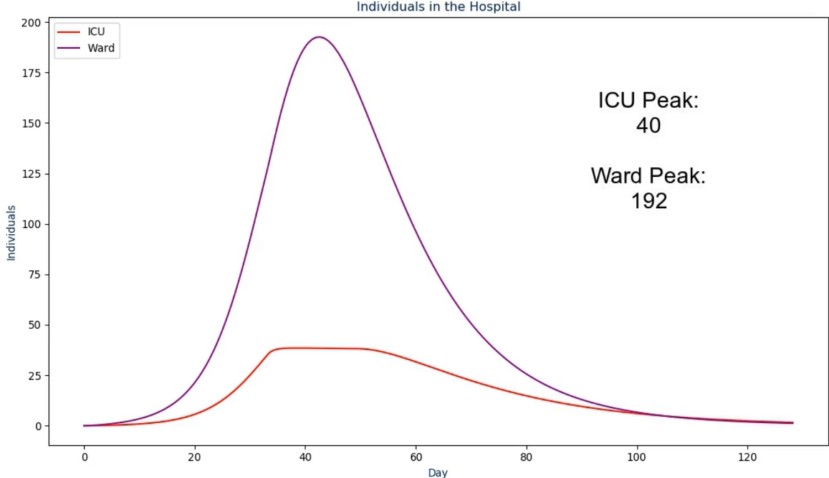

**Figure 3    The third frame in the proposed solution.**

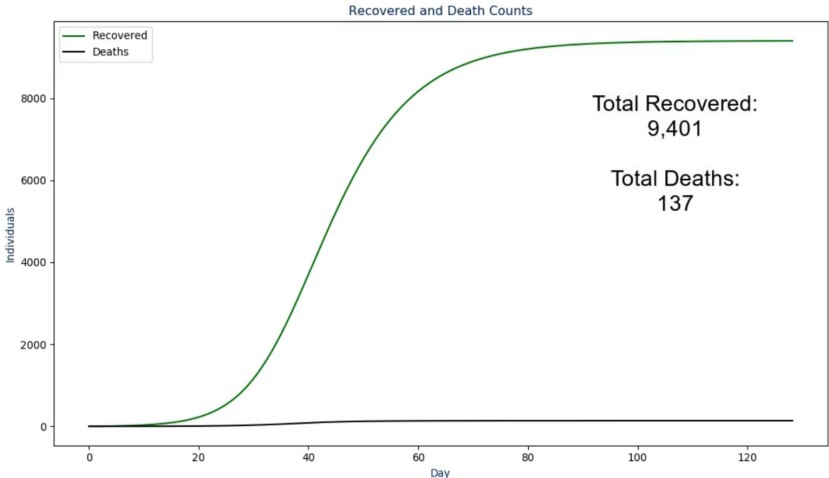

**Figure 4    The fourth frame in the proposed solution.**

was created in Python and uses the libraries matplotlib (*Hunter, 2007*) along with moviepy (https://zulko.github.io/moviepy/) to generate the result discussed.

### Audio output

To further enhance the video output, the choice was made to have a text-to-speech library interpret the results for the viewer. However, a simple, robotic voice was not desirable as it may sound too monotonous for the context and not feel trustworthy as a result. Therefore, this work uses an artificial intelligence assisted solution from *Tachibana, Uenoyama & Aihara (2018)*. To summarize, this novel approach uses deep convolution networks to adjust the actual audio spectrograms from the text-to-speech to make it sounds more realistic and less robotic. The authors made this codebase publically available along with a

pre-trained model that is used in this work. In the introductory slide (Fig. 1), the following text was converted to audio using this library:

```
Visualization of NorthCOVID-19. The results in this video are based on the
following parameters: an initial population of "N", initial infected "N_i",
ICU capacity of "N_c", contact rate of "c", an infectivity of "tau" percent,
and an illness duration of "nu" days
```

The parameters in this text and the ones to follow use the notation from NorthCOVID-19 (*Savage et al., 2020*). Since this output is a generated dynamically, the text is updated based on these values. In the next frame (Fig. 2) the following text is spoken:

```
This scenario started with "N_i" infected individuals. Once the infection
ended after "INFECTION_DURATION" days, there was a total of
"TOTAL_INFECTED_COUNT" individuals that had become infected
```

Where ''INFECTION_DURATION'' and ''TOTAL_INFECTED_COUNT'' come from their respective values but as the **absolute** values instead of per capita. In the next frame (Fig. 3) the following text is spoken:

```
Over the course of the infection, the ICU capacity peaked at "ICU_PEAK_COUNT"
individuals while the Ward peaked at "WARD_PEAK_COUNT" individuals. The ICU
limit in this scenario was set to "N_c" individuals
```

Where ''ICU_PEAK_COUNT'' and ''WARD_PEAK_COUNT'' come from the maximum value in their respective sections of the model. In the next frame (Fig. 4) the following text is spoken:

```
Out of the "TOTAL_INFECTED_COUNT" individuals that become infected, a total
of "TOTAL_RECOVERED_COUNT" individuals recovered from the infection while
"TOTAL_DEATH_COUNT" deaths were recorded
```

Where ''TOTAL_RECOVERED_COUNT'' and ''TOTAL_DEATH_COUNT'' come from their respective values but as the **absolute** values instead of per capita. Lastly, in the conclusion slide (Fig. 5) the following text is spoken:

```
After the infection ended in "INFECTION_DURATION" days. The total infected
peaked at "TOTAL_INFECTED_COUNT", the ICU peaked at "ICU_PEAK_COUNT", the
Ward peaked at "WARD_PEAK_COUNT", the recovered peaked at
"TOTAL_RECOVERED_COUNT", and the deaths peaked at "TOTAL_DEATH_COUNT"
```

By having this audio accompany the video output, the goal is to assist the user in their understanding of the results in a simple, to the point, format. The result is an informative 2 min video (with audio) that, from start to finish, takes around 10 min to generate.

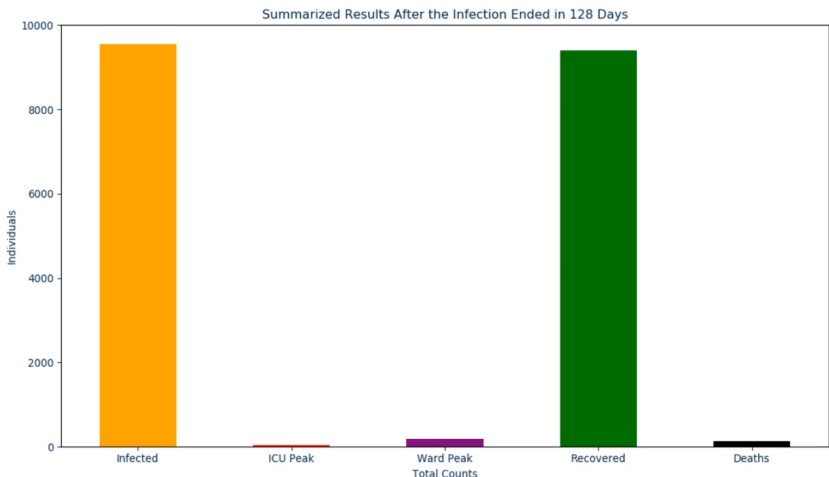

**Figure 5**  The final frame in the proposed solution.

## RESULTS

### User study

To evaluate the web interface of the NorthCOVID-19 model and the proposed video output, a 15 question survey was created. The link to this form was posted on the authors' Linkedin (total of 38 individuals responded) and Slack channels (total of two individuals responded) with the only restriction being that it could be filled out only once. When an individual was first presented with the survey, they were shown a 4-minute video that walked them through how to use the web interface that was provided by the authors of *Savage et al. (2020)*. Then, they were asked to use the platform on their own and answer 10 questions that were adapted from the System Usability Scale (SUS) (*Brooke, 1996*) as per Figs. 6A–6J. An answer of 1 meant they "Strongly Disagree" with the statement and 5 meant "Strongly Agree". Lastly, they were shown a sample output of the proposed video (https://www.youtube.com/watch?v=ZhUw6zJRnFE) in this article, and asked 5 questions regarding their thoughts about it as per Figs. 7A–7E.

### Survey

After completing the survey, a total of 40 individuals responded. Since the first ten questions (Figs. 6A through 6J) were directly adapted from SUS (*Brooke, 1996*), a score was derived to see how simple the interface is to use. Once these values were calculated for each individual response, the average was determined to get the final result where 0 means that the interface is not usable at all and 100 means that it is extremely easy to use. From studies done on this survey, a score of **68** is noted as being average for this measure (*Bangor, Kortum & Miller, 2008*; *Lewis, 2018*). This survey conducted on NorthCOVID-19 scored the interface at **70.94** which is slightly above the average. This result, along with the issues raised show the need to create a different way of interfacing with this model. In analyzing these results further (Figs. 7A through 7E), it can be seen that a majority of the responses leaned towards a confident understanding of the video output and improved comprehension of the results.

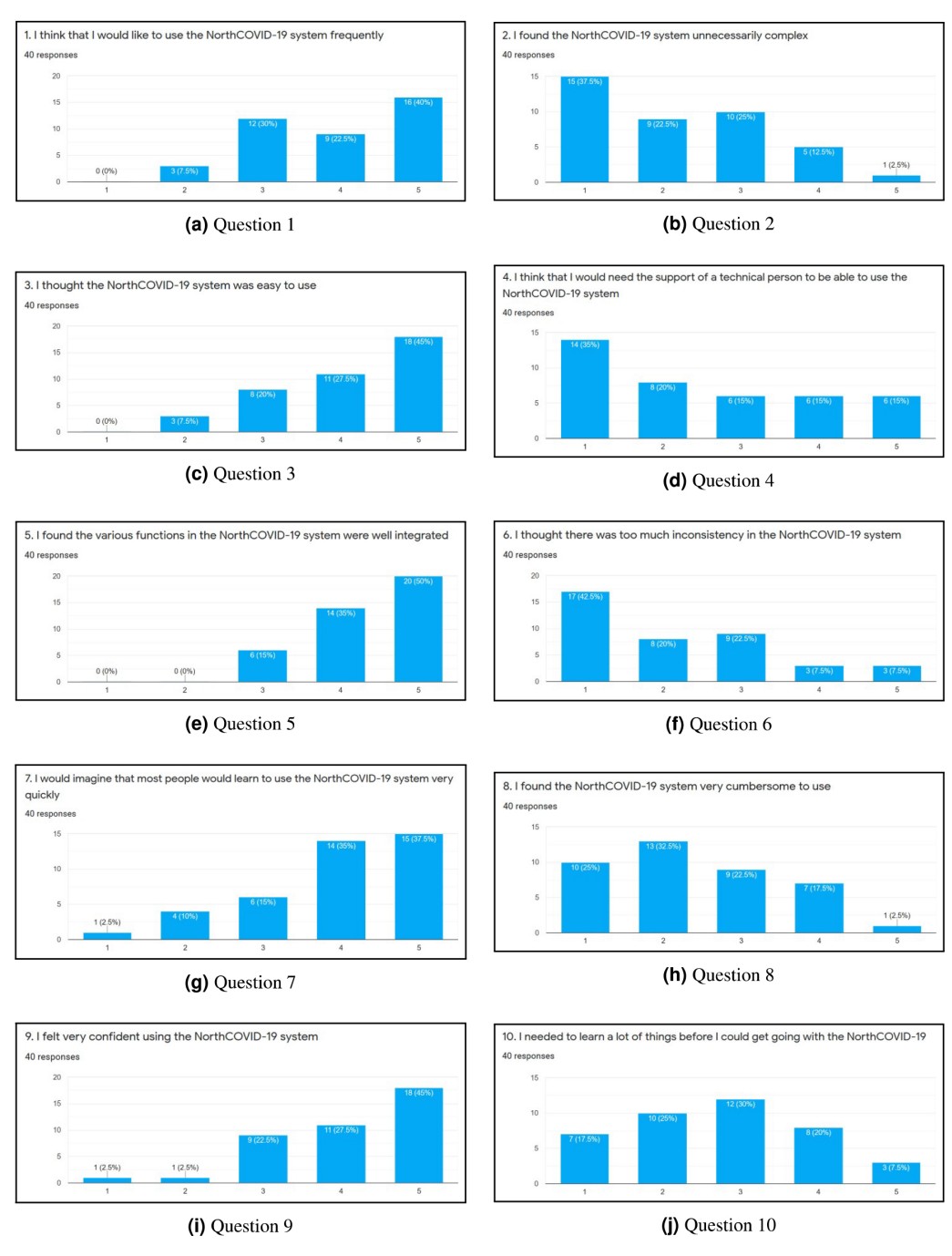

**Figure 6** (A-J) The results of each question in the first part of the survey.

An interesting observation, however, can be seen in Fig. 7B) when the individuals were asked if the website output was easier to understand. Most answers fell under a neutral response (*i.e.,* score of 3) but enough fell under scores 4 and 5 (*i.e.,* "Agree" and "Strongly Agree") that it leaned more towards agreeing than disagreeing with the statement. This response could be attributed to the fact that the website is entirely interactive with its

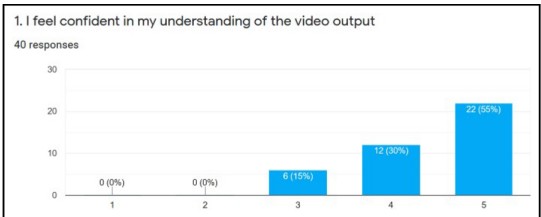

**(a)** Question 1

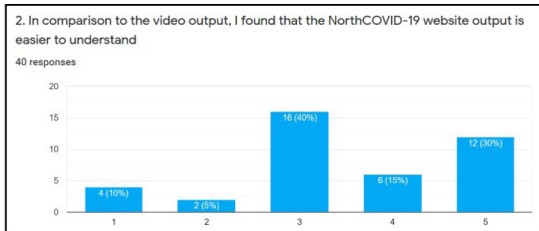

**(b)** Question 2

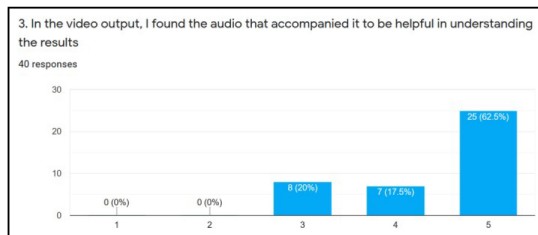

**(c)** Question 3

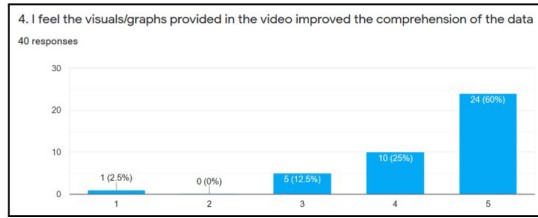

**(d)** Question 4

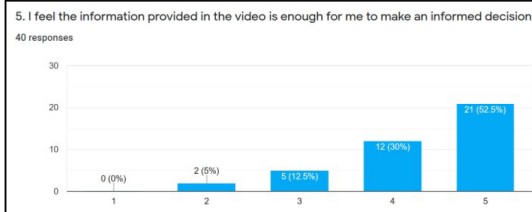

**(e)** Question 5

**Figure 7** **(A-E) The results of each question in the second part of the survey.**

graphs and transparency in regards to the parameter configuration. Regardless, from Figs. 7A, 7C, 7D, and 7E a strong leaning towards the video output being easy to understand, improving comprehension, and being helpful to the individuals who took the survey can be observed in these results. This shows achievement in the goal of meeting ACM codes 1.3, 2.7, 3.1, and 3.2 as discussed before.

## LIMITATIONS

There are three limitations that we encountered with this study. First, although the responses to the survey seem promising, we do acknowledge that the collection technique used is a limitation in itself. We had decided against recording demographic and educational information of the respondents to ensure anonymity, but can see how that could've provided further insight to these ratings. Second, the number of individuals that responded was relatively low, and is therefore another limitation of this work. Lastly, we have not considered optimizing the video generation process which would be ideal for a real-time implementation as it takes up to 10 min for the script to generate roughly 7,200 frames. We welcome more feedback on this visualization and hope to continue improving its output so that all can understand.

## CONCLUSIONS

In this article, the outputs of an epidemic model called "NorthCOVID-19" (*Savage et al., 2020*) along with its web interface was examined to see what issues it may have when being interpreted by the general public. The interface was created with researchers in the field in mind but the issues affects even the general population regardless of their background. After identifying three main issue that could cause anxiety and confusion, this article focused on creating a solution that would thoroughly explain the results in an ethical, easy to understand format. The result was a video that eases in to each of the results while having an artificial intelligence assisted voice further explaining the proposed output the viewer is seeing. This feature was designed with the ACM code of ethics (*Gotterbarn et al., 2017*) in mind as we wanted to ensure that the output presented was easy for the general public to understand (*Kienzler, 1997*; *Pickering & Kara, 2017*). To evaluate the NorthCOVID-19 interface, questions were adapted from the System Usability Scale (SUS) (*Brooke, 1996*) to get a score out of 100 that describes how easy it is to use. The resulting score was calculated to be 70.94 which is slightly above average (*Bangor, Kortum & Miller, 2008*; *Lewis, 2018*). When users were asked about the video output in comparison to the website's output, the results showed most answers being neutral with a leaning towards the website's output being easier to understand. This could be attributed to the interactive graphs provided and transparency of the parameter configuration. However, when discussing the video output itself, the results strongly leaned towards it helping improve the comprehension of the results and being easy to understand. This shows that the proposed output does add value to NorthCOVID-19 and that the design choices involved(having an animated video output with audio) achieved their intended purpose of providing the user with the information needed to understand the results without an interface. We have opted to

make the simulation model, video generation script, and survey results open-sourced at the following repository: https://github.com/andrfish/NorthCOVID19.

### Funding
Andrew Fisher received funding from SSHRC Insight grant 21820. The cost of publication and other expenses are covered by NSERC Discovery Grant, held by Dr. Vijay Mago. The funders had no role in study design, data collection and analysis, decision to publish, or preparation of the manuscript.

### Grant Disclosures
The following grant information was disclosed by the authors:
SSHRC Insight grant 21820.
NSERC Discovery Grant.

### Competing Interests
The authors declare there are no competing interests.

### Author Contributions
- Andrew Fisher conceived and designed the experiments, performed the experiments, analyzed the data, performed the computation work, prepared figures and/or tables, authored or reviewed drafts of the paper, and approved the final draft.
- Neelkumar Patel performed the experiments, analyzed the data, performed the computation work, prepared figures and/or tables, and approved the final draft.
- Preetkumar Patel performed the experiments, analyzed the data, performed the computation work, prepared figures and/or tables, and approved the final draft.
- Pruthvi Patel performed the experiments, analyzed the data, performed the computation work, prepared figures and/or tables, and approved the final draft.
- Vinit Krishnankutty performed the experiments, authored or reviewed drafts of the paper, and approved the final draft.
- Vaibhav Bhat performed the experiments, authored or reviewed drafts of the paper, and approved the final draft.
- Parth Valani performed the experiments, authored or reviewed drafts of the paper, and approved the final draft.
- Vijay Mago analyzed the data, authored or reviewed drafts of the paper, and approved the final draft.
- Abhijit Rao analyzed the data, authored or reviewed drafts of the paper, and approved the final draft.

### Data Availability
The data is available at OSF: Fisher, Andrew. 2022. ''An Ethical Visualization of the NorthCOVID-19 Model.'' OSF. April 28. osf.io/j8erc.

The code is available at GitHub: https://github.com/andrfish/NorthCOVID19.

## Supplemental Information

Supplemental information for this article can be found online at http://dx.doi.org/10.7717/peerj-cs.980#supplemental-information.

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
