# Peer review of "An ethical visualization of the NorthCOVID-19 model"

_PeerJ Computer Science, doi:10.7717/peerj-cs.980_

## Round 0.1 · original submission · Major Revisions

Your paper has been reviewed by the experts. They have major concerns to be addressed. therefore, you are requested to carefully look into the reviewer's concerns and carefully incorporate them before resubmission.

·

Basic reporting

I’m impressed how many words can be spent for explaining and justifying the design of a simplified user interface. The paper reads very well, however, I can’t really assess its value as the videos themselves, not to speak of the system generating the videos on the fly, are not available, in contrast to the original website, https://covid.datalab.science, for which the simplified user interface has been developed for.

I suggest to add a few words explaining how the outcome of the reported work will be used/made available in the future.

Experimental design

no comment

Validity of the findings

Can't be assessed as the main result of the paper (a video-generating add-on to a given web interface) is not available.

Additional comments

This paper has nine authors; it could be helpful to see how each of the authors contributed to the reported work.

There are no raw data.

minor:
line 119: there seems to be something missing after ‘section’;

·

Basic reporting

The authors examine the outputs of an epidemic model called “NorthCOVID-19” along with its web interface to see what issues it may have when being interpreted by the general public. They identified three main issues that could cause anxiety and confusion and focused on creating a solution that would thoroughly explain the results in an ethical, easy-to-understand format. They built audio and video methods including using AI to communicate the epidemic model results.

This is a well-written paper that I recommend to be accepted.

Experimental design

The experimental design is clear with sufficient details to reproduce the experiments and results.

Validity of the findings

Conclusions are well-stated.

Reviewer 3 ·

Basic reporting

Overall it looks interesting.
However, I am not too sure if the assumptions made are true.

Experimental design

Assuming that the proposed model is for public use. I don't think so. You cannot find the model anymore.

Validity of the findings

One needs to provide some facts about how much the model was used by the general public.
How independent the study can be if the first author of this paper is the second author of the published model paper?
A sample of 40 is not too bad, but not knowing who the participants are in real trouble.

---

## Round 0.2 · Minor Revisions

There are few more corrections needed, please revise and resubmit.

·

Basic reporting

() re individual contributions of all authors: I assume that this will be visible in the final paper; so far this information is not included.

() https://github.com/andrfish/NorthCOVID19, Video Generation Script for the COVID Crushers,
introductory comment:
“This script takes input from NorthCOVID-19 (see model here) and produces a video animation (see sample_format.pdf) of the results. The file "sample_output.csv" is an example of what the output will look like, "sample_parameters.json" is an example saved file of the parameters used, and "sample_results.json" is an example of the results output to the website from the simulation.”

-> none of the links work - ‘NorthCOVID-19’ , ‘model’;
for some of the pointers there are no links given at all, eg. sample_format.pdf, sample_results.json, sample_parameters.json.

() “The result was a video”
thanks for adding the video generation feature to the original website https://covid.datalab.science; I suggest to move the link to this website, hidden so far in a footnote, to a more prominent position.

() “Please do not leave this page, the generation may take up to 10 minutes and the download will start automatically.”
-> having to wait up to 10’ (it actually took almost as long) is a rather long delay for getting something simplified compared to what I immediately get and comprises actually more information.

() the last slide in the generated “video” (is more a sequence of slides) needs some normalisation to make the results visible.

in summary,
I’m still amazed how many words one can spend for explaining and justifying the design of a simplified user interface. I personally would prefer the direct output as offered by the original website https://covid.datalab.science; not to say that I somehow feel offended that I - as a user- is considered not to be able to understand what the original diagrams are telling me.

But I’m happy to accept that I might not belong to the actual target group of the tool.

Experimental design

N/A

Validity of the findings

N/A

Additional comments

none

---

## Round 0.3 · accepted · Accept

Thanks for addressing the comments.

---

## Author Rebuttal · Round 0.3

**DATE:**  April 20, 2022

**TO:**  Academic Editor, PeerJ Computer Science

**FROM:**  Andrew Fisher, Department of Computer Science
Lakehead University, Thunder Bay, ON, Canada P7B 5E1

**RE:**  Article resubmission.

Dear Editor,

We wish to submit a revised copy of our article entitled " An ethical visualization of the NorthCOVID-19 model" for consideration by PeerJ Computer Science. We can attest that this is an original article that has not been published elsewhere, nor is it currently under submission for publication elsewhere.

We have carefully reviewed the concerns and believe they were addressed with an update of our open-source repository as well as minor changes to the manuscript (highlighted in blue). Please see our point-by-point response in the pages following this cover letter.

We have no conflicts to disclose.

Please address all correspondence concerning this article to me at afisher3@lakeheadu.ca.

Thank you for taking the time to consider this article.

Sincerely,

Andrew Fisher
Research Assistant,
Department of Computer Science
Lakehead University, Thunder Bay, ON
Tel:204-761-9726
Web:http://www.datalab.science

Reviewer 1 (Monika Heiner)
Basic reporting
() re individual contributions of all authors: I assume that this will be visible in the final paper; so far this information is not included.
Yes, the author contribution breakdown was provided in the PeerJ submission itself.

() https://github.com/andrfish/NorthCOVID19, Video Generation Script for the COVID Crushers, introductory comment:
"This script takes input from NorthCOVID-19 (see model here) and produces a video animation (see sample_format.pdf) of the results. The file "sample_output.csv" is an example of what the output will look like, "sample_parameters.json" is an example saved file of the parameters used, and "sample_results.json" is an example of the results output to the website from the simulation."

-> none of the links work - 'NorthCOVID-19' , 'model';
for some of the pointers there are no links given at all, eg. sample_format.pdf, sample_results.json, sample_parameters.json.
Thank you for pointing this out, we have corrected the links for "NorthCOVID-19" and "model". For the files, we have now included them in our repository (https://github.com/andrfish/NorthCOVID19).

() "The result was a video"
thanks for adding the video generation feature to the original website https://covid.datalab.science; I suggest to move the link to this website, hidden so far in a footnote, to a more prominent position.
We have moved this link in-line towards the top of page 2.

() "Please do not leave this page, the generation may take up to 10 minutes and the download will start automatically."
-> having to wait up to 10' (it actually took almost as long) is a rather long delay for getting something simplified compared to what I immediately get and comprises actually more information.
For the video generation process, each slide is first rendered and saved based on their respective statistics using the OpenCV and Matplotlib libraries. Then, each slide is combined and rendered again to get the final result that the user receives. We opted for it to play back at 60 frames per second to give smooth animations, for a total of roughly 7,200 frames to render in up to 10 minutes. A real-time implementation wasn't considered for this study so we have added this optimization constraint to our limitations towards the top of page 11.

() the last slide in the generated "video" (is more a sequence of slides) needs some normalisation to make the results visible.
We appreciate this feedback but felt that normalizing it would add another complexity for the general public to understand rather than displaying the actual values. This is now emphasized towards the end of the "Issues" subsection on page 4.

in summary,

I'm still amazed how many words one can spend for explaining and justifying the design of a simplified user interface. I personally would prefer the direct output as offered by the original website https://covid.datalab.science; not to say that I somehow feel offended that I - as a user- is considered not to be able to understand what the original diagrams are telling me.

But I'm happy to accept that I might not belong to the actual target group of the tool.

Thank you for your understanding, we certainly did not intend for the output to be directed towards researchers. Rather, it is for the general public which we hope is clear to readers from our "Public Understanding" and "Issues" sections.